# Mitochondrial Proteomes in Neural Cells: A Systematic Review

**DOI:** 10.3390/biom13111638

**Published:** 2023-11-11

**Authors:** Aya Nusir, Patricia Sinclair, Nadine Kabbani

**Affiliations:** 1Interdisciplinary Program in Neuroscience, School of Systems Biology, George Mason University, Fairfax, VA 22030, USA; anusir@gmu.edu; 2School of Systems Biology, George Mason University, Fairfax, VA 22030, USA; psincla2@gmu.edu

**Keywords:** mass spectrometry, energetics, protein quantification, bioinformatics, neurodegeneration, evolution, synapses

## Abstract

Mitochondria are ancient endosymbiotic double membrane organelles that support a wide range of eukaryotic cell functions through energy, metabolism, and cellular control. There are over 1000 known proteins that either reside within the mitochondria or are transiently associated with it. These mitochondrial proteins represent a functional subcellular protein network (mtProteome) that is encoded by mitochondrial and nuclear genomes and significantly varies between cell types and conditions. In neurons, the high metabolic demand and differential energy requirements at the synapses are met by specific modifications to the mtProteome, resulting in alterations in the expression and functional properties of the proteins involved in energy production and quality control, including fission and fusion. The composition of mtProteomes also impacts the localization of mitochondria in axons and dendrites with a growing number of neurodegenerative diseases associated with changes in mitochondrial proteins. This review summarizes the findings on the composition and properties of mtProteomes important for mitochondrial energy production, calcium and lipid signaling, and quality control in neural cells. We highlight strategies in mass spectrometry (MS) proteomic analysis of mtProteomes from cultured cells and tissue. The research into mtProteome composition and function provides opportunities in biomarker discovery and drug development for the treatment of metabolic and neurodegenerative disease.

## 1. Introduction

Mitochondria are essential organelles that evolved via a symbiotic relationship between early prokaryotes and primitive eukaryotic cells over a billion years ago [1,2,3]. Commonly known as the powerhouse of the cell, mitochondria are key players in many cell functions, including energy-producing oxidative phosphorylation (OXPHOS), apoptosis, and intracellular calcium and lipid management [3,4]. Mitochondrial activity is critical for cellular protein synthesis and degradation that may be perturbed in various human diseases.

Structurally, mitochondria are double-membraned organelles that contain two compartments: an intermembrane space (IMS) and a matrix, formed by an outer mitochondrial membrane (OMM) and an inner mitochondrial membrane (IMM) [3] (Figure 1A). The IMS possesses the machinery for the translocation of proteins through the mitochondria, while the matrix contains mitochondrial DNA (mtDNA) and the machinery required for its transcription and translation. The IMM and mitochondrial ribosomes (mtRibosomes) interact with nucleoid complexes consisting of mtDNA bound to core and peripheral region proteins [5,6]. The core region mitochondrial proteins such as transcription factor A (TFAM), Twinkle (TWNKL), RNA polymerase (POLRMT), single-stranded DNA binding protein (mtSSB), and polymerase subunit gamma (POLG) are involved in the replication and transcription of mtDNA [7]. The peripheral region proteins include ATPase family AAA domain containing protein 3 (ATAD3) and prohibitin (PHB1 and PHB2), which interact with TFAM and are involved in nucleoid organization and stability [6,7].

The OMM is in contact with various intracellular organelles including the nucleus and the endoplasmic reticulum (ER) [8,9]. ER-mitochondrial contact sites, also known as mitochondria-associated membranes (MAMs), are critical for the exchange of calcium and lipids [10,11,12]. The IMM folds in a manner that enables the formation of cristae projections into the matrix, thereby increasing the surface area for ATP production [13,14]. Abnormalities in cristae morphology can result in the accumulation of reactive oxygen species (ROS) within mitochondria and are associated with various neurodiseases [15,16]. Both the IMM and OMM participate in the transfer of proteins important for mitochondrial homeostasis, including fission and fusion, excursion of mitochondrial-derived vesicles (MDVs), and mitophagy [17,18,19].

Mitochondria are highly heterogeneous, exhibiting differences in shape, size, mtDNA copy number, membrane potential (ψ_m_), and metabolic properties [20,21,22,23,24,25,26]. Proteomic studies also reveal significant variations in the composition of mtProteomes in various cell and tissue types. For example, approximately 15% of the mtProteome is altered amongst astrocytes, granule cells, and Purkinje cells in the cerebellum [27]. Heterogeneity is also observed when comparing mtProteomes from synaptic and non-synaptic neuronal compartments [28,29]. It is suggested that almost half of the proteins found within the mtProteome can participate in human disease, especially disorders of the nervous system [30].

In this review, we examine how advancements in mass spectrometry (MS) and the development of bioinformatic tools and proteome compendiums contribute to understanding mtProteome properties in neural cells. We examine the functional classification of proteins that make up mtProteomes and discuss how protein heterogeneity can impact mitochondrial regulation and disease states with an emphasis on metabolic and neurodegenerative disorders.

## 2. Proteins Involved in Mitochondrial Quality Control

Mitochondrial quality control is a homeostatic adaptive process involving the regulation of mitochondrial fission and fusion, derivation of MDVs, and mitophagy. Changes in mitochondrial-damage-associated molecular patterns (mtDAMPs) are often seen in cell inflammatory states, with specific mitochondrial components designated as mtDAMPs, including mtDNA, ROS, ATP, cardiolipin (CL), cytochrome c (CyC), and calcium ions [31,32,33] (Figure 2). Many mitochondrial proteases also participate in protein turnover, which is crucial in maintaining healthy mitochondrial functioning and quality control. For example, several proteases such as Lon peptidase 1 (LONP1) and AAA proteases (i.e., YME1L and m-AAA) are involved in vital functions including protein degradation, folding, and transport, as well as mtDNA maintenance [34,35,36,37]. Mitochondrial quality control is an important process for the management of cellular stress responses that arise during nutrient deprivation, toxin exposure, and other environmental triggers.

### 2.1. Fission and Fusion

The studies show that mitochondria display network features within cells that are modified by changes in mitochondrial motility, as well as fission and fusion, which guide their spatial distribution in cells. In neurons, mitochondria exhibit varied morphologies and densities in synaptic compartments relative to the rest of the cell [29,38]. Changes in mitochondrial networks are coupled to fission and fusion and can be rapidly modified to match the energetic demands of the cells. In general, increased fission results in small and rounded mitochondria, while greater fusion favors the formation of thin elongated mitochondria [39,40,41,42]. Adaptations in fission support mitochondrial quality by enabling the segregation of impaired mitochondrial components, while fusion can compensate for mitochondrial deficits (i.e., metabolic shifts) by rapidly enabling mitochondrial expansion and ATP production [18,43]. Various mtProteome components including optic atrophy 1 (OPA1) and mitofusin (MFN1 and MFN2) drive fusion, and dynamin-related protein 1 (DRP1) and fission protein 1 (FIS1) support fission [44].

### 2.2. Mitochondrial-Derived Vesicles (MDVs) and Mitophagy

The mitochondrial life cycle begins with biogenesis and ends with degradation through mitophagy (a form of autophagy). In some cases, however, quality control can mitigate acute mitochondrial damage via the excursion of MDVs that carry damaged components to degradative lysosomes [45]. This form of degradation is unlike mitophagy, which involves the destruction of the entire organelle. The studies indicate that MDVs act as the first line of defense as they can rapidly remove marked cargo, whereas mitophagy is a last resort [38,46]. Both MDVs and mitophagy are important quality check points for eliminating aged or damaged mitochondrial components [45,47]. MDVs are also heterogeneous vesicles expressing diverse surface protein markers that are involved in directing their degradation pathway. MDV heterogeneity is also contingent upon its cargo and stimuli. For example, single-membrane MDVs are marked with the expression of proteins such as translocase of the OMM 20 (TOMM), which binds to pathway-dependent tether proteins to initiate lysosomal degradation [19,32,45,48]. Double-membrane MDVs, on the other hand, do not contain TOMM20 but are positive for matrix cargo such as pyruvate dehydrogenase (PDH) [48]. MDVs can also contain mitochondrial antigens that trigger immune responses in macrophages and dendritic cells via sorting nexin-9 (SNX9) and Ras-related protein dependent processes during immune regulation [49,50,51].

Mitophagy is triggered in response to mitochondrial damage and the activation of mitophagy receptors, driving the expansion of a phagophore around mitochondria [52]. Currently, a number of key mitophagy pathways have been identified in neural cells, including stress-induced mitophagy, Parkin-mediated mitophagy, and amyloid-beta- and tau-induced mitophagy [52]. As one of the more studied pathways, Parkin-mediated mitophagy appears linked to neurodegeneration during Parkinson’s disease (PD) [53]. In this pathway, a stress signal activates PTEN-induced putative kinase 1 (PINK1), a mitochondrial and cytosolic kinase protein, causing a mitochondrial signaling cascade through the phosphorylation of Parkin (ubiquitin E3 ligase) and ubiquitination of mitochondrial substrates (mtSubstates) at the surface of the OMM [54]. These events recruit optineurin, interacting with microtubule-associated protein light chain 3 (LC3) to initiate autophagy and lysosomal degradation of mitochondria [55,56]. PINK1 and Parkin can also regulate the motility and fusion of mitochondria and direct the formation of MDVs under various conditions [44].

## 3. Specialized Features of Neuronal Mitochondria

The properties of neuronal mitochondria are critical for neuronal signaling, metabolism, healthy neurotransmission, and synaptic plasticity [57,58]. Mitochondrial morphology, dynamics (e.g., fission), and subcellular distribution are related to the state of the cell, with considerable differences between developing and adult neurons. For example, as synaptic activity increases during development and metabolic demand rises, synaptic mitochondrial density increases [59,60,61,62]. Additionally, the formation of synapses is highly dependent on the localization of mitochondria to sites of growth (e.g., growth cones), and as energy demands increase, mitochondrial transport increases, enabling mitochondrial delivery across elaborate cell structures [61,63,64]. Mitochondrial metabolic activity appears to vary across neuronal segments, with axonal structures such as non-myelinated nodes, branch points, and presynaptic terminals containing a greater cluster of anchored mitochondria [29,38,65,66]. Mitochondria located at distal axonal segments and presynaptic terminals, however, are at a higher risk for accumulating damage and can contribute to the emergence of synaptic disease [67,68] (Figure 3). This is in part because mitochondrial biogenesis and degradation (e.g., mitophagy) occur mostly near the soma and depend on axonal transport [69].

### 3.1. Cytoskeletal Interactions Involved in Mitochondrial Transport

Mitochondria are trafficked to and from sites of neurotransmission including dendritic spines and presynaptic terminals through interactions with cytoskeletal proteins conferring both directionality and the rate of mitochondrial transport. The movement of mitochondria along the axon is regulated by microtubule transport, whereas mitochondrial localization involves actin binding [70]. Mitochondria may become anchored to endocytic vesicles, the plasma membrane, and the ER as their motility appears to decrease following synaptic maturation [38,71,72,73].

The axonal cytoskeleton participates in mitochondrial transition between motile and static states depending on neuronal energy demands. The anterograde and retrograde movement of mitochondria along axons is driven by interactions with kinesin and dynein microtubule motor proteins, respectively [70] (Figure 3). Various mtProteome components participate in cytoskeletal transport by linking the mitochondria to motor proteins. For example, a family of adaptors known as Milton/TRAK proteins were found to interact with Miro, an OMM protein, forming complexes that connect kinesin and dynein motors with the mitochondria [74,75]. Compromised interactions between mitochondria and the cytoskeleton under various experimental conditions are shown to disrupt synaptic plasticity and neurotransmitter release, and can contribute to synaptic protein accumulation, a hallmark of neurodegenerative disease [76,77].

### 3.2. Mitochondrial Proteins Involved in Lipid Transport and Regulation

Mitochondrial proteins participate in intracellular lipid trafficking, ensuring the exchange of various types of lipids between mitochondria and other organelles, including the plasma membrane [78]. Lipid trafficking is also vital for mitochondrial membrane integrity and can modulate energy production during shifts in metabolic demands [79]. The exact percentage of the mtProteome involved in lipid regulation can vary depending on the criteria used to define lipid regulation. It can also differ according to cell type or developmental state since the rates and modes of lipid transport across mitochondria drastically differ. A substantial number of mitochondrial proteins participate in lipid metabolism and regulation, with many of these proteins providing support for mitochondrial membrane integrity (e.g., CL), energy production (e.g., steroidogenic acute regulatory protein (StAR), and various forms of lipid signaling.

Mitochondrial lipid trafficking through the OMM enables lipid exchange with other compartments such as the ER. One key protein is the mitochondrial translocator protein (TSPO), which interacts with lipid-binding proteins to facilitate the bidirectional transport of cholesterol across the OMM [80]. Mitochondrial channels such as voltage-dependent anion channels (VDAC) also transfer lipids across the OMM and in and out of the matrix [81]. The IMS serves as an important intermediate compartment for mitochondrial lipids with Ups family proteins (Ups1 and Ups2) involved in the movement of phospholipids between the OMM and the IMM [82]. The transfer of key phospholipids such as CL and phosphatidylethanolamine (PE) across mitochondrial membranes is critical during mitochondrial biogenesis [83].

The IMM is rich in CL, a unique mitochondrial phospholipid vital for the structural integrity of the IMM and shown to impact OXPHOS-mediated ATP production [84,85]. Proteins in the IMM, including tafazzin, play an essential role in CL synthesis [86]. The mitochondria also contain enzymes that are important for fatty acid breakdown through β-oxidation. One such enzyme is carnitine palmitoyltransferase I (CPT), which is located on the OMM. CPTI transfers fatty acids into the mitochondrial matrix during β-oxidation, resulting in the generation of acetyl-CoA for the tricarboxylic acid cycle (TCA) and ATP production [87].

The increasing research on MAM contact sites is enabled by the biochemical analysis of MAM isolation and the characterization of MAM-associated proteins that are essential for the exchanges between ER and mitochondria. A strong number of proteins have been identified within MAMs including: MFN2, involved in phosphatidylserine (PS) transfer from the ER into mitochondrial membranes; calcium uniporter (MCU), which is essential for calcium uptake into mitochondria; regulator of microtubule dynamics protein 3 (RMDN3), which is important for mitochondrial trafficking; inositol trisphosphate receptor (IP_3_R); acetyl-CoA transporter (SLC25A1); and StAR [88,89,90].

## 4. Strategies for the Study of mtProteomes

The protocols for mitochondrial isolation are widely available [91,92]. Here, we present an overview of common strategies amongst the existing protocols.

### 4.1. Mitochondrial Homogenization, Isolation, and Solubilization

Mitochondrial isolation is an important first step in investigating the mtProteome and for insight into conditions that can modify mitochondrial processes. Mitochondrial isolation can also be used to perform functional measures of mitochondrial activity ex vivo (Figure 4). Depending on the study design, the starting material (tissue or cells) must be harvested in relatively high quantities and may require pooling many samples in order to obtain sufficient mitochondrial quantity (1–5 × 10^9^ cells) [91]. First, the samples are washed and incubated with hypotonic buffers that destabilize membranes. This is followed by physical homogenization in a dounce homogenizer to disrupt cell membranes, releasing contents while preserving mitochondrial integrity [93].

Cellular debris and other components can be separated from the mitochondria either by centrifugation or affinity purification. Centrifugation is the most common method for separating mitochondria from other cellular organelle through density gradients and/or differential centrifugation. Sucrose, Ficoll, or Percoll gradients can effectively separate subcellular components based on their relative density and sizes. Differential centrifugation, on the other hand, uses varying centrifugation forces (speeds and duration) to separate components by mass. Both separation techniques can successfully isolate mitochondria from cells and tissue sources.

Immunoaffinity isolation has been used to isolate mitochondria by genetic expression of an epitope tag such as 3× hemagglutinin (HA) Mito-tag to a mitochondrial protein within cultured cells or in transgenic mice [94,95]. These tags have also been expressed conditionally to examine cell-specific responses in glutamatergic neurons of a mouse model of Leigh Syndrome, a disorder associated with mitochondrial dysfunction [94]. Antibodies against the epitope tag can be adsorbed to magnetic beads and used to isolate mitochondria from the sample solution [96]. In studies that do not use a genetic tag, immunoaffinity isolation can still be performed depending on the affinity and specificity of an antibody for an endogenous mitochondrial protein. OMM proteins such as TOMM22, for example, have been successfully used to obtain mitochondrial fractions from brain tissue [27].

Assessing the isolated mitochondrial fraction for content purity is accomplished by Western blot detection of a mitochondrial protein such as CyC [97]. It is not uncommon that even the most efficient isolation can yield some minor non-mitochondrial proteins. These contaminants may occur through mitochondrial-associated sites such as MAMs [98]. However, depending on the experiment, such a result can also be used to assess protein interactions between mitochondria and other organelles, including the cytoskeleton and the ER within the mitochondrial fraction [44].

Mitochondria are highly heterogeneous, exhibiting complex mtProteome compositions depending on tissue and cell-type, as well as subcellular compartment. Thus, adapting an appropriate isolation technique to the question of interest is an important experimental consideration in a proteomic study. In cases where mitochondria are to be used for physiological assays, additional isolation measures are required to ensure mitochondrial ψ_m_ and ATP production. Global mtProteome profiling, on the other hand, necessitates that most mitochondrial proteins are efficiently solubilized from membranes for proteomic detection.

### 4.2. Mass Spectrometry and Bioinformatic Analysis

MS-based proteomic analysis is a rapidly evolving strategy useful for quantifying proteins and post-translational modifications under various experimental conditions [99,100]. Mitochondrial fractions are prepared for MS analysis by standard sequential steps, including precipitation, denaturing, and enzymatic cleavage using trypsin or chymotrypsin to generate cryptic peptide sequences for MS analysis based on mass-to-charge ratio (*m*/*z*). Software packages such as MaxQuant v2.4.10.1 and Proteome Discoverer 3.1 can be used after the MS run in order to compare the acquired to the predicted spectra within databases such as NCBI or UniProt [101]. The relative quantification of each protein can be performed using a label-free or labeled proteomic approach [102,103]. In label-free MS, each sample is analyzed independently, and thus, label-free quantification (LFQ) is based either on the number of peptide-spectral matches or ratiometric comparison of the area under the curve for peptides associated with a particular protein. Label-free methods can be used to identify and quantify many proteins with minimal preparation time. However, LFQ is limited by the potential for high variability between samples since each is run through the instrument separately. MS quantification can also be performed by the addition of heavy or light isotopes (e.g., ^15^N or ^12^C) that are incorporated into proteins during synthesis and, thus, can be detected by MS based on the altered *m*/*z* measure. Isobaric labels such as tandem mass tags (TMT) or isobaric tags for relative or absolute quantification (iTRAQ) are added to the protein sample during sample preparation. Isobaric tag-based quantification is assessed by analyzing reporter ions that are released during the high-energy collision dissociation step. This labeling method allows for “multiplexing” or combining samples from different experimental conditions into a single MS run, and thus, each run can assess protein content from all experimental conditions. This process substantially reduces the time required and can be useful in overcoming the variability between runs that is seen during label-free MS analysis.

The data can be bioinformatically analyzed using various online tools including the Database for Annotation, Visualization and Integrated Discovery (DAVID) and Search Tool for the Retrieval of Interacting Genes/Proteins (STRING), which are useful for enrichment analyses using gene ontology (GO) and other pathway databases [104,105,106]. An enrichment analysis can be used to compare the experimentally derived list of proteins against a background list of proteins and to determine the statistical significance [104]. MitoCarta3.0 is a leading catalog of mitochondrial proteins curated by the Broad Institute (MIT/Harvard) with information on mitochondrial localization and pathway participation based on 1136 currently known mitochondrial proteins [107], as described below.

## 5. A Snapshot of the mtProteome

A single mitochondrion may contain up to 10 copies of its 16 kb circular genome, with each encoding 13 proteins, 22 tRNAs, and 2 rRNAs [108,109] (Figure 1B). The 13 proteins make up approximately 1% of the mtProteome, with the remaining 99% coming from proteins that are delivered to the mitochondria [110]. Mitochondrially encoded proteins are crucial for mitochondrial activity, forming core units of the OXPHOS system for ATP production. Nuclear genes, on the other hand, encode various transport proteins, assembly factors, and mitochondrial respiratory chain structural subunits. Over the last two decades, the proteomic analysis of mitochondria has revealed a mosaic of protein groups that make up the mtProteome, which can be represented by protein abundance or protein group function and localization. The MitoCarta3.0 inventory provides annotations of submitochondrial localization and functional pathways from 1136 proteins across mitochondria from 14 human tissue samples [107]. Interestingly, less than half of all mitochondrial proteins within MitoCarta3.0 are found to be shared between the 14 tissue types, and many mitochondrial proteins appear only in specific tissue. Our survey of 811 mitochondrial proteins from human neural tissue (cerebrum, cerebellum, brainstem, and spinal cord) in MitoCarta3.0 is discussed.

### 5.1. mtProteome Analysis in MitoCarta3.0

The localization of proteins into mitochondria requires cellular targeting mechanisms since the majority of such proteins are imported from the cytosol [110]. Submitochondrial targeting must also ensure that proteins are delivered to the right mitochondrial compartment for their proper function. The data from MitoCarta3.0 suggest that almost half of the neural mitochondrial proteins reside within the matrix, with the rest distributed between the IMM, IMS, and OMM compartments (Figure 5A).

Proteins involved in metabolism are found to make up over a third of the mtProteome dataset (Figure 5B) and this includes proteins involved in the breakdown of carbohydrates, lipids, and amino acids. The second highest category is the “mitochondrial central dogma” associated with the regulation of the mitochondrial genome (mtGenome). Specifically, these proteins support DNA, RNA, and tRNA processes, as well as mtRibosome production and activity. OXPHOS proteins, which include complex subunits involved in protein assembly, stability, and maturation, make up another significant portion, with a lower representation of proteins involved in mitochondrial morphology and quality control (e.g., mitophagy). A recent bioinformatic analysis of >8000 proteins in mitochondrial preparations of human cells defined a mtProteome consisting of >1100 proteins [30]. This mtProteome analysis suggests that the two most abundant mitochondrial proteins are the heat shock proteins 60 and 10 (HSP), which are important chaperones for protein folding and quality control. Interestingly, these HSPs are found as components of the “protein import, sorting and homeostasis” category that constitutes only ~8% mitochondrial proteins within MitoCarta3.0 (Figure 5B).

### 5.2. mtProteome Heterogeneity

The changes in the proteins that make up mtProteomes relate important metabolic differences between brain cells such as neurons and glia, and heterogeneity within mtProteomes can further reflect the adaptive responses of mitochondria to local energy requirements, as shown for the higher energy demand at synapses. In neuroimmune cells such as macrophages and microglia, a dynamic shift to an altered energy system (from OXPHOS to glycolysis) accompanies immune cell activation [111,112,113]. An analysis of mtProteomes may serve to identify mitochondrial modifications such as shape, metabolic activity (e.g., ATP production), and may support identifying factors that lead to disease states.

#### 5.2.1. mtProteome Heterogeneity within Neural Cells

The analysis of mtProteomes in neurons and astrocytes revealed that the fusion protein MFN1 is enriched in astrocyte mitochondria, whereas the fission protein DRP1 is higher in neuronal mitochondria [114]. These findings are in agreement with earlier studies that show that astrocytes contain a greater proportion of elongated mitochondria that undergo more fusion, while neurons have smaller and rounded mitochondria with more fission activity [115,116]. The studies also suggest that astrocytes are more likely than neurons to utilize glycolysis in energy production [117,118]. However, astrocytes play an important role in supporting neuronal metabolism and may directly supply mitochondria to neurons under select conditions [119]. Beyond their supportive role, the evidence indicates that astrocyte mitochondria are more efficient at metabolizing long-chain fatty acids and various types of lipids. This is consistent with findings that astrocyte mtProteomes are enriched in fatty acid oxidation enzymes such as ATP citrate lysate, acetyl-CoA dehydrogenase short chain (ACADS), CPT1a and CPT2, and peroxisomal proteins [27,114]. These enrichments in astrocyte mtProteomes underscore the notion that these glial cells are critical for brain lipid homeostasis.

Astrocytes also provide important antioxidant support to local neuronal networks, and neuronal antioxidant defense and neuroplasticity are highly coupled to interactions between neurons and astrocytes. Interestingly, the transplantation of astrocyte mitochondria into neurons in vitro appears to mediate oxidative stress and promote neuroplasticity in neurons following injury, suggesting that mitochondrial activity is critical for antioxidant management [120]. The proteomic analysis of astrocyte mitochondria shows an enrichment of glutathione peroxidases 1 and 4 (GPX1 and GPX4), peroxiredoxin 6 (PRDX6), microsomal glutathione s-transferase 1 (MGST1), as well as coproporphyrinogen oxidase (CPOX) proteins [114,121]. Conversely, in neuronal mitochondria, different classes of antioxidant proteins are enriched, including microsomal glutathione s-transferase 3 (MGST3), peroxiredoxin 2 (PRDX2), and superoxide dismutase type 1 (SOD1) [114]. Mutations in SOD1 protein have been implicated in various proteinopathies such as amyotrophic lateral sclerosis (ALS) and PD, due to its suggested role in regulating neuronal apoptosis [122,123,124,125,126].

Calcium buffering in mitochondria is mediated by various protein complexes and transporters that differ in expression between neural cells. Using the example of the cerebellum, the mtProteome of Purkinje cells exhibits significant enrichment in RMDN3, which localizes to MAMs and participates in ER calcium release [27]. In comparison, the calcium buffering properties of granule cells appear more highly dependent on MCU function, and MCU proteins are found to be higher in granule cells [27].

#### 5.2.2. mtProteome Heterogeneity between Synaptic and Non-Synaptic Compartments

Synaptic activity is highly dependent on the localization and activity of mitochondrial proteins. Mitochondria localized to synapses are found to exhibit different morphologies and protein expression patterns than non-synaptic ones. Studies indicate that ~400 mitochondrial proteins can be differentially expressed between synaptic and non-synaptic sites, almost 40% of which are associated with disorders of the nervous system [28,29]. The analysis shows that heterogeneity between synaptic and non-synaptic mtProteomes occurs in the expression of important functional proteins, including those directly involved in OXPHOS, fission/fusion, calcium transport, and mtDNA replication [28]. The expression of all ETC proteins except CyC oxidase subunit 4 isoform 2 (COX412) appear reduced within synaptic mitochondria, resulting in decreased overall ATP production [28]. The studies also show that a reduction in ETC protein expression reduces the mitochondrial respiration rate and can predispose synapses to oxidative stress properties that may contribute to neurodegenerative disease [127]. Mitochondrial fragmentation, induced by higher levels of fission proteins (e.g., DRP1) and lower levels of fusion proteins (e.g., OPA1, MFN1, and MFN2), is also common in synapses and can contribute to a reduction in ATP production by synaptic mitochondria [28,43].

Synaptic mitochondria, thus, present a unique damage risk based on the high energy demand of the presynaptic terminal and the reduction in ATP energy production capacity. Additionally, mitochondria at the presynaptic terminals of longer axons are subject to slower trafficking and turnover [71]. It has been shown that such presynaptic mitochondria exhibit higher ROS production, more calcium sensitivity, and greater damage accumulation over time relative to mitochondria localized to other neuronal compartments [67,128]. Calcium regulation is mediated by MCU and mitochondrial calcium uptake 1 (MICU1), which appear to be reduced in synaptic mitochondria [28]. Although MCU knockdown has been shown to attenuate calcium toxicity in mice hippocampal and cortical neurons, the downregulation of both MCU and MICU1 in synaptic mitochondria may be contributing to the increased vulnerability to excitotoxicity, as MICU1 exhibits neuroprotective effects [129,130].

TFAM proteins are necessary for mtDNA transcription and translation, participating in the protection of mtDNA [7,14,131]. The proteomic analysis of neuronal cultures from mice indicates a reduction in TFAM in synaptic mitochondria, suggesting a role in higher sensitivity to mtDNA damage at the synapses [28,132].

## 6. The mtProteome in Health and Disease

The studies indicate that over half of all human genes encoding mitochondrial proteins may be impacted during human disease [30]. Mutations in the genes that make up mtProteomes, whether mitochondrial or nuclear of origin, may contribute to neurodegenerative disorders such as Alzheimer’s disease (AD). Specifically, when mutations arise in mtDNA, the impact seems to be localized to ETC activity, however, mutations in nuclear-encoded mitochondrial proteins appear to present a broader effect on mitochondrial function [133].

### 6.1. Inherited mtDNA Disease

Inherited mtDNA diseases are a group of rare genetic disorders caused by mutations in the mtGenome. Some examples include:Mutations in mtDNA or nuclear genes encoding OXPHOS proteins involved in the ETC, including complexes I, II, III, IV, and V (Table 1).Proteins involved in the TCA cycle including citrate synthase and succinate dehydrogenase.Mutations in genes involved in CoQ10 biosynthesis, resulting in CoQ10 deficiency.Mutations in the nuclear gene POLG, resulting in defective mtDNA replication and mitochondrial DNA deletions [134].

These disorders principally affect the function of mitochondria yet manifest with a range of clinical symptoms. Inherited mtDNA disease exhibits a maternal inheritance pattern with mtDNA mutations often coexisting with normal mtDNA in the same cell. This phenomenon, called heteroplasmy, confers an effect on disease severity depending on the ratio of mutant to normal mtDNA in cells. Inherited mtDNA diseases are often associated with symptoms such as muscle weakness, neurological deficits, cardiac abnormalities, and developmental delays [134]. In the brain, inherited mtDNA disease can significantly impact neural function, resulting in seizures and mitochondrial encephalopathy with lactic acidosis and stroke-like episodes (MELAS) [135].

### 6.2. Metabolic Dysfunction

Metabolic dysfunction, specifically in the ETC, not only directly affects energy production but also results in ROS accumulation that is associated with Huntington’s disease (HD) and other neurodegenerative conditions [136,137]. Mutations in subunits of complex I are linked to LS, specifically NDUFS4, and this depletion of ETC subunits drastically reduces ATP respiration [94]. In AD, mitochondrial hypometabolism seems to occur as a result of changes in ETC function, glucose uptake by neurons, and changes in astrocyte metabolic activity [138,139,140]. The recent evidence shows that AD may result in an altered mtProteome profile, since changes in the expression of ETC complex proteins such as ATP-synthase have been observed in patients [139]. Quantitative proteomic profiling of mtProteomes from early-onset AD patients indicates a reduction in the expression of complex I NADH dehydrogenase subunits NDUFA1, NDUFA2, NDUFA5, NDUFA9, NDUFA10, NDUFB3, NDUFB6, NDUFB8, NDUFB11, NDUFS4, NDUFS7, and NDUFAB1 [139]. The mtProteome of late-onset AD patients also shows a reduction in the expression of NDUFA3, NDUFA7, NDUFB4, NDUFS1, and NDUFS3, as well as CyC oxidases such as COX5A, COX5B, COX7A2, COX7A2L, and CYC1 [139]. The downregulation of ETC complexes has also been implicated in autism spectrum disorders, where ATP5A1 and ATP5G3 (Complex V), as well as NDUFA5 (Complex I), were found to be significantly reduced in all brain regions of autistic patients [141]. Conversely, an increase in the expression of ETC proteins, such as subunits of complexes I-V, is seen in the brains of HD patients [142].

Mutations in SOD1 are associated with apoptosis in motor neurons during ALS, and the proteomic analysis supports a downregulation in the expression of SOD1 during ALS [114,122,123,124,125,143]. ALS is also marked by changes in several other mitochondrial proteins, including PHB1 and PHB2, involved in maintaining mtDNA stability and quality control [6,7,143,144].

Age-dependent changes in HSPs, specifically HSP60 expression, were suggested to confer vulnerability to neurodisease through mitochondrial regulation. This is supported by experiments in a mouse model of PD, showing a reduction in HSP60 levels within the substantia nigra, resulting in protein aggregation within neurons during aging [145]. As previously discussed, HSP60 and HSP10, proteins that are involved in the regulation of protein folding, are amongst the most abundant proteins in the human mtProteome [30].

### 6.3. Disorders of Mitochondrial Quality Control and Transport

Imbalanced mitophagy and fission/fusion can contribute to neurodegenerative diseases including AD, PD, and HD. The studies indicate that disruptions in PINK1/Parkin signaling can reduce mitophagy, resulting in an accumulation of damaged mitochondria within neurons [56,146,147]. Protein interactions between PINK1/Parkin and those involved in fission/fusion participate in the regulation of mitochondrial dynamics, which, when disrupted, can signal apoptosis [148]. The overexpression of PINK1/Parkin was shown to directly increase mitochondrial fission, while its inactivation increases fusion [149]. Increasing the expression of DRP1 can rescue PINK1/Parkin deficits and restore fission/fusion balance [150]. Furthermore, PINK1 is shown to influence cristae morphology by interacting with mitochondrial IMM proteins, including (MIC60), a stabilizer of cristae [151]. As with other PINK1 interactors, the overexpression of MIC60 was found to neutralize PINK1 dysfunction and protect cells from damage in vitro [151,152].

The aggregation of amyloid proteins, such as amyloid-beta and tau proteins within neurons, can increase the expression of DRP1 and FIS1, resulting in hyperfission (Figure 2) [153,154]. Similarly, an increase in fission and a decrease in fusion were observed within primary neurons in mouse models of AD, suggesting a role for altered proteins including OPA1, MFN1, and MFN2 [153]. The disruption of mitochondrial quality control, specifically fission and fusion, appears to be common in various neurodegenerative disorders, and mutations in OPA1 and MFN2 are also linked to autosomal dominant optic atrophy and Charcot–Marie–Tooth neuropathy, respectively [155,156]. Mutant huntingtin (mHTT) aggregates, found in mitochondrial matrices, are shown to interact with various quality control proteins that alter fission/fusion events including DRP1, FIS1, MFN1, MFN2, and OPA1 [142]. Mutations in DRP1 also result in excessive fusion, leading to the degeneration of Purkinje neurons of the cerebellum in mice [157].

Studies have shown that MDVs contribute to the release of mitochondrial proteins in extracellular vesicles, which may be disrupted by damage or various disease states [32]. This transfer of mitochondrial contents (i.e., mtDNA) and membrane proteins (i.e., COX2) between cells is shown to contribute to changes in metabolic activity and tumor growth [32,158,159,160]. For instance, mtDAMPs released into the extracellular space induce the release of neuroinflammatory cytokines during neurodegenerative states [32]. Similarly, a rise in CyC within the cytosol signals apoptosis and neurotoxicity that can contribute to neurodegeneration [161,162].

Bidirectional axonal transport of mitochondria between the synaptic terminal and soma are required for efficient ATP production and mitophagy. Impairment in mitochondrial transport is, thus, an important driver of neurodegenerative states and shown to contribute to AD, PD, and HD synaptic pathology [163]. In particular, the reduced expression of mitochondrial transport proteins (i.e., Miro1, syntaphilin, TRAK1, and TRAK2) is associated with increased amyloid precursor protein (APP) accumulation in a mouse model of the Swedish APP mutation [164].

### 6.4. Involvement of MAMs in Calcium Excitotoxity and Neurodegenerative Disease

Mitochondrial dysfunction, specifically during high ROS conditions (e.g., during high calcium load), increases mitochondrial permeability transition pore (mtPTP) formation [165]. This process has been observed in AD and appears to be driven by an increase in cyclophilin D (CyPD) expression [166]. CyPD is a mitochondrial chaperone critical for mtPTP formation that is activated under high cytosolic calcium levels [167]. Inhibiting CyPD may be sufficient in protecting against mtPTP-associated calcium toxicity [166].

MAMs are highly important for regulating the entry of calcium into mitochondria, and proteomic analysis has identified MAM proteins that participate in neuronal damage [168]. Mutations in IP_3_R, which are enriched in MAMs, were shown to contribute to spinocerebellar ataxias type 2 and 3 [169,170]. The MCU complex within MAMs is also differentially expressed across cerebellar cell types and confers cell-specific calcium sensitivity [27]. Targeted efforts have focused on MCU to block excitotoxicity in neurons. In one example, MCU knockdown was shown to attenuate calcium toxicity in hippocampal and cortical neurons in mice [171]. Mutation in AD-related genes, such as presenilin (PS1 and PS2), can also disrupt calcium homeostasis through MAMs, and alterations in interactions between PS1/PS2 and IP_3_R were shown to contribute to cell death in human lymphoblasts and mouse cortical neurons [172,173,174,175,176].

Beyond calcium buffering, structural and functional disruptions in MAMs and their components may also contribute to neurodegeneration. Several MAM proteins, such as protein kinsase-like endoplasmid reticulum kinase (PERK) and MFN2, are implicated in AD pathogenesis [177]. Similarly, ALS, PD, and HD are also characterized by mutations or changes in expression of various MAM-associated proteins such as Sig-1R, Parkin, and htt, respectively [178].

## 7. Limitations and Future Directions

Advancements in proteomic technologies are expanding our knowledge on the composition and regulatory properties of mtProteomes. The current efforts, however, are not without limitations that must be overcome for a more complete understanding of how mitochondrial proteins participate in cell function. Some of these limitations include:Managing the inherent problem of mitochondrial heterogeneity within experiments and data analysis by considering factors such as variability in mitochondrial size, shape, and genomic composition.The overwhelming majority of mitochondrial proteins are synthesized in the cytosol and imported into mitochondria. Thus, many mitochondrial proteins are not always localized within mitochondria, and studies of mtProteomes are limited in information on such dynamics.A subset of mitochondrial proteins is present in relatively low abundance yet may participate in important processes. Efforts will need to be made to continue to improve on protein capture and detection.Genetic complexity and heteroplasmy further contribute to challenges in the diagnosis of mitochondrial diseases and require increased sensitivity in genomic detection methods for mtDNA mutations.

## Figures and Tables

**Figure 1 biomolecules-13-01638-f001:**
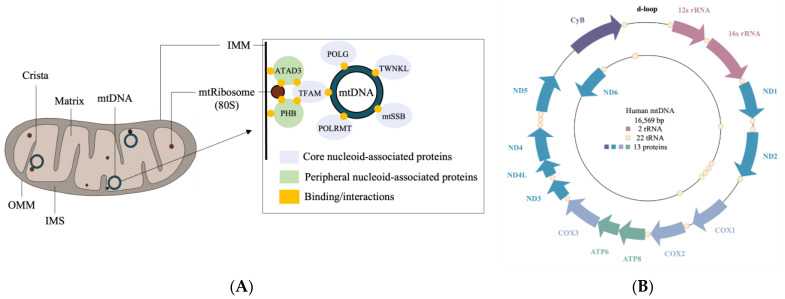
The structure of and genomic properties of mitochondria. (**A**) Mitochondrial compartments include the outer mitochondrial membrane (OMM); the intermembrane space (IMS); the inner mitochondrial membrane (IMM), which folds to form cristae; and a matrix that houses mitochondrial DNA (mtDNA) and 80S ribosomes (mtRibosomes). Inset: mtDNA binds core proteins that interact with peripheral proteins, mtRibosomes, and the IMM, forming a nucleoid. Core proteins include transcription factor A (TFAM), Twinkle (TWNKL), RNA polymerase (POLRMT), single-stranded DNA binding protein (mtSSB), and polymerase subunit gamma (POLG), whereas peripheral proteins include ATPase family AAA domain containing protein 3 (ATAD3) and prohibitin (PHB1 and PHB2). (**B**) Human mtDNA (16,569 bp) encodes 2 rRNA, 22 tRNA, and 13 proteins that form subunits of oxidative phosphorylation (OXPHOS) complexes, including NADH dehydrogenases (ND1, ND2, ND3, ND4, ND4L, ND5, and ND6), cytochrome c oxidases (COX1, COX2, and COX3), ATP synthases (ATP6 and ATP8), and cytochrome b (CyB).

**Figure 2 biomolecules-13-01638-f002:**
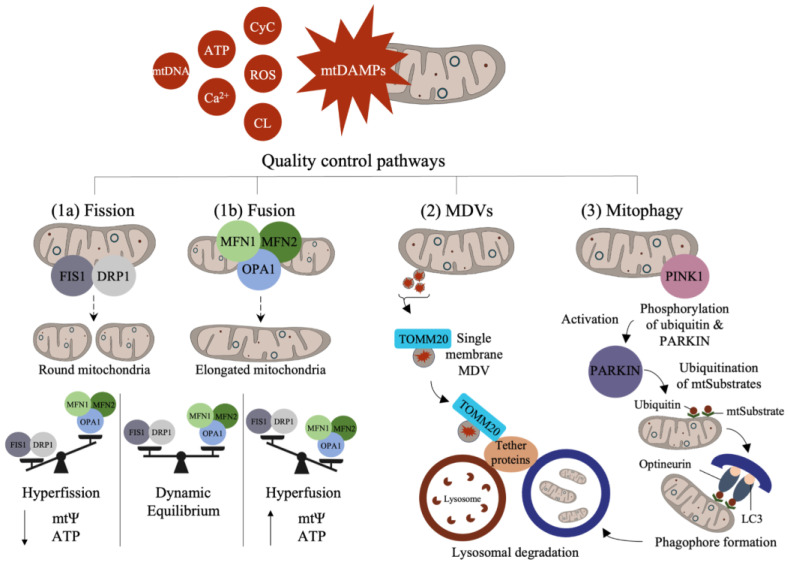
Adaptive properties in mitochondrial quality control. Mitochondrial-damage associated molecular patterns (mtDAMPs) appear in inflammatory states including high concentrations of mtDNA, ATP, calcium ions (Ca^2+^), cytochrome c (CyC), reactive oxygen species (ROS), and cardiolipin (CL). Mitochondria react to mtDAMPs via fission, fusion, derivation of mitochondrial-derived vesicles (MDVs), and mitophagy. Fission is mediated by fission 1 (FIS1) and dynamin-related protein 1 (DRP1), forming rounder mitochondria. Fusion is regulated by mitofusin (MFN1 and MFN2) and optic atrophy 1 (OPA1), forming elongated mitochondria. (1a–b) Fission and fusion exist in a dynamic equilibrium during healthy states: hyperfission (increased FIS1 and DRP1) is associated with decreases in mitochondrial membrane potential (mtΨ) and ATP production; hyperfusion is associated with increased mtΨ and ATP. (2) Single membrane MDVs express translocase of the outer mitochondrial membrane 20 (TOMM20), carrying damaged components for lysosomal degradation. (3) Mitophagy degradation through various pathways such as Parkin-mediated mitophagy, which begins with the phosphorylation of Parkin by PTEN-induced putative kinase 1 (PINK1) and the ubiquitination of mitochondrial substrates (mtSubstates), triggering optineurin interaction with microtubule-associated protein light chain 3 (LC3) and leading to the formation of a phagophore and mitophagosome that fuses with the lysosomes.

**Figure 3 biomolecules-13-01638-f003:**
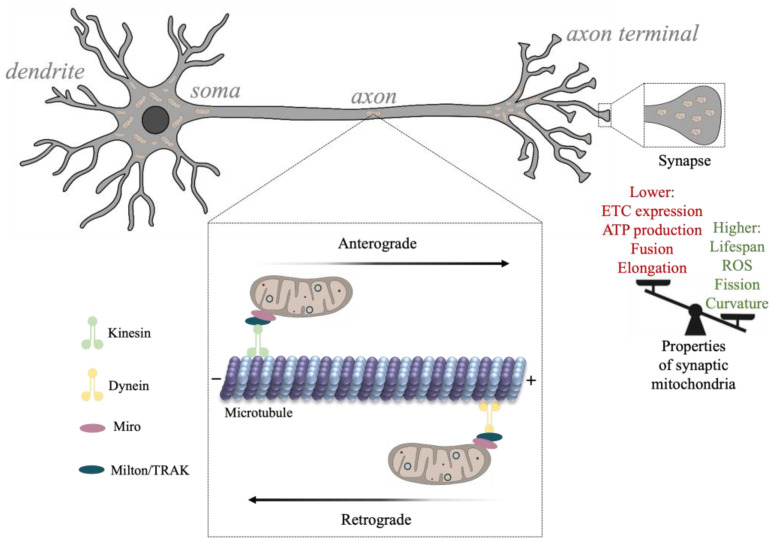
The transport and function of synaptic mitochondria. Movement of mitochondria along the axonal cytoskeleton is mediated by kinesin and dynein microtubule motors that are complexed with Milton/TRAK proteins and Miro, a protein localized to the outer membrane of mitochondria. Kinesin traffics mitochondria from soma to axon terminals (anterogradely) and dynein back to the soma (retrogradely). As neurons develop, the rate of mitochondrial trafficking to synapses decreases, hence, synapses are dense in mitochondria and accommodate high energy demands at presynaptic terminals. Synaptic mitochondria exhibit unique properties including higher levels of ROS, fission, and lower ATP production.

**Figure 4 biomolecules-13-01638-f004:**
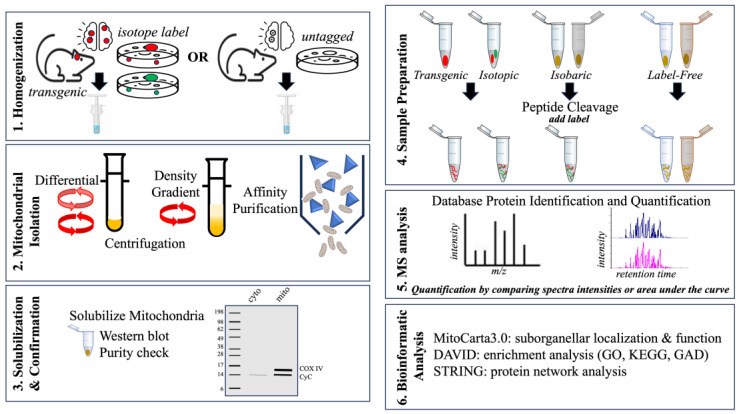
Summary of mtProteome analysis. (1) Mitochondria isolated from mouse models or cultured cells. (2) Mitochondrial extraction obtained using centrifugation and/or density gradient separation. Mitochondria can also be isolated using tagged affinity purification (e.g., HA-Mito-tag). (3) Mitochondrial fraction purity can be confirmed using a Western blot detection of mitochondrial proteins (e.g., CyC). (4) Protein samples can be prepared for MS analysis using internal tagging (e.g., isotopic or isobaric) or label-free quantitative approaches. (5) MS analysis using methods such as electrospray ionization (ESI) liquid chromatography (LC) enables peptide separation and detection. (6) Protein results can be further analyzed using bioinformatic tools such as STRING network analysis.

**Figure 5 biomolecules-13-01638-f005:**
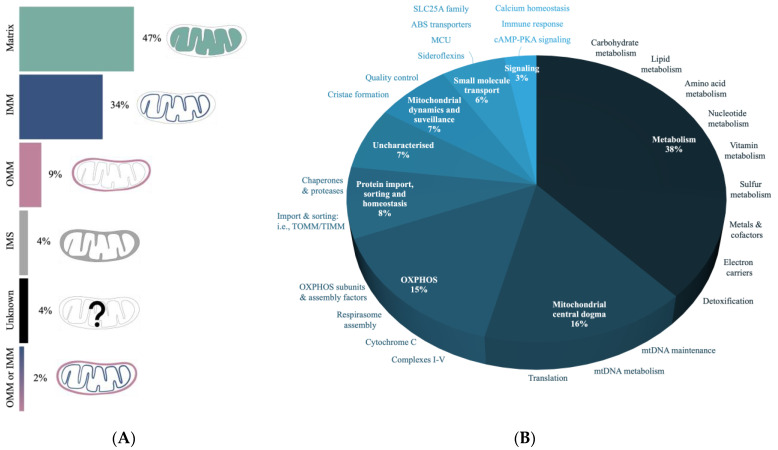
Annotation of mtProteomes in CNS using MitoCarta3.0. (**A**) The submitochondrial distribution profiles of the mtProteome based on compartments: matrix, inner mitochondrial membrane (IMM), outer mitochondrial membrane (OMM), and intermembrane space (IMS). (**B**) Characterization of mtProteome-based functional properties. Most proteins are involved in various types of metabolism, followed by the mtDNA regulation and oxidative phosphorylation (OXPHOS).

**Table 1 biomolecules-13-01638-t001:** Proteins encoded by mtDNA.

mtGene	Protein	Complex
ND1	Subunit 1 of NADH dehydrogenase	I
ND2	Subunit 2 of NADH dehydrogenase	I
ND3	Subunit 3 of NADH dehydrogenase	I
ND4	Subunit 4 of NADH dehydrogenase	I
ND4L	Subunit 4L of NADH dehydrogenase	I
ND5	Subunit 5 of NADH dehydrogenase	I
ND6	Subunit 6 of NADH dehydrogenase	I
CyB	Cytochrome b	III
COX1	Subunit 1 of cytochrome c oxidase	IV
COX2	Subunit 2 of cytochrome c oxidase	IV
COX3	Subunit 3 of cytochrome c oxidase	IV
ATP6	Subunit 6 of ATP synthase	V
ATP8	Subunit 8 of ATP synthase	V

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
