# Peer review of "Mitochondrial Proteomes in Neural Cells: A Systematic Review"

_biomolecules, 2023, doi:10.3390/biom13111638_

Round 1

Reviewer 1 Report

Comments and Suggestions for Authors

In this manuscript titled “Mitochondrial Proteomes in Neural Cells: A Systematic Review”, Aya Nusir et al discuss the recent progress in the composition and properties of mtProteomes important for mitochondrial energy production, quality control, and lipid signaling in neural cells. Moreover, the authors highlight the important role of The mtProteome in health and disease. The contents of this manuscript are potentially interesting to the readers, and the manuscript is well written.

Major comments

1. in the section of “2. Proteins Involved in Mitochondrial Quality Control”, the authors can supplement some mitochondrial proteases including LONP1, OMA1, Yme1L, m-AAA etc since they play important role in the degradation or processing of mitochondrial proteins.

2. in the section of “7. Limitations and Future Directions”, the limitations and challenges about mitochondrial diseases should be supplemented.

3. In Figure 2, mtDAMPs should contain mtDNA, the related contents can be supplemented.

Reviewer 2 Report

Comments and Suggestions for Authors

Nusir et al., presented a review paper entitled “Mitochondrial proteomes in neural cells: a systematic review”. In this paper, after a brief introduction on mitochondria structure organization and function, the authors have summarized mitochondrial proteins involved in mitochondrial quality control but also proteins involved in mitochondrial transport and lipid transport and regulation. Authors also point out the strategies to study mitochondrial proteome and highlighted mitochondrial proteome heterogeneity between different neural cells (neurons versus glial cells) or between synaptic (enriched in mitochondria and highly dependent on their activity) and non-synaptic compartments. Finally, the authors reviewed mitochondrial proteomes alteration in diseases such as inherited mitochondrial DNA disease or neurodegenerative diseases. The paper was well-written, however, addressing the below questions will increase the accuracy of the review paper and would be extremely useful for the inexperienced reader seeking to gain an accurate understanding of the status of knowledge in the complex field and the most salient scientific contributions.                                                       

Major suggestions:

·         In the 2.2. section, the authors have described mitochondrial-derived vesicles (MDVs) expressing diverse surface protein markers but highlighted only single membrane MDVs expressing TOMM20 marker. Is it possible to describe other MDVs subtypes with their origin and distinctive cargoes? For review: DOI: 10.1016/j.biocel.2016.07.020.

·         Moreover, the authors have described MDVs as the first line of defense as they can transport mitochondrial damaged components to the late endosomes or multivesicular bodies (MVBs) in order to be degraded by the lysosomes. However, MBVs can be also redirected towards and fuse with the plasma membrane allowing the secretion of mitochondrial components in extracellular vesicles. As a consequence, it would be nice to highlight studies describing extracellular vesicles from mitochondria origin and/or containing mitochondria as well their potential involvement in neurodegenerative diseases in 6.3. section. For exemple: DOI: 10.1038/s41467-021-21984-w; DOI: 10.1038/s41593-019-0486-0; DOI: 10.1126/sciadv.abe5085; DOI: 10.1038/nature18928 .

·         In “The mtProteome in Health and Disease” part, it would be nice to add a section “alteration of mitochondrial transport machinery”. For exemple: DOI: DOI: 10.21203/rs.3.rs-3235215/v1.

·         The involvement of MAMs was also largely explored in neurodegenerative diseases. For review: DOI: 10.3390/ijms21249521; DOI: 10.1007/s12035-016-0140-8; DOI: 10.3390/cells10071789.

Minor suggestions:

Line 485: To change the title 6.2. “Inherited mtDNA Disease” which was probably copy paste from 6.1.

Round 2

Reviewer 2 Report

Comments and Suggestions for Authors

After a first review, the authors have collected new data according to the comments of reviewers. We appreciate the authors efforts and works to answer all the comments and suggestions, and we agree for the publication.

Sincerely,